

# Emotion detection from handwriting and drawing samples using an attention-based transformer model

Zohaib Ahmad Khan[1], Yuanqing Xia[1], Khursheed Aurangzeb[2], Fiza Khaliq[3], Mahmood Alam[4], Javed Ali Khan[5] and Muhammad Shahid Anwar[6]

[1] School of Automation, Beijing Institute of Technology, Beijing, China
[2] Department of Computer Engineering, College of Computer and Information Sciences, King Saud University, Riyadh, Saudi Arabia
[3] School of Computer Science and Technology, Beijing Institute of Technology, Beijing, China
[4] School of Computer Science and Engineering, Central South University, Hunan, China
[5] Department of Computer Science, School of Physics, Engineering, and Computer Science, University of Hertfordshire, Hatfield, United Kingdom
[6] Department of AI and Software, Gachon University, Seongnam-si, South Korea

Corresponding author
Yuanqing Xia,
xia_yuanqing@bit.edu.cn

## ABSTRACT

Emotion detection (ED) involves the identification and understanding of an individual's emotional state through various cues such as facial expressions, voice tones, physiological changes, and behavioral patterns. In this context, behavioral analysis is employed to observe actions and behaviors for emotional interpretation. This work specifically employs behavioral metrics like drawing and handwriting to determine a person's emotional state, recognizing these actions as physical functions integrating motor and cognitive processes. The study proposes an attention-based transformer model as an innovative approach to identify emotions from handwriting and drawing samples, thereby advancing the capabilities of ED into the domains of fine motor skills and artistic expression. The initial data obtained provides a set of points that correspond to the handwriting or drawing strokes. Each stroke point is subsequently delivered to the attention-based transformer model, which embeds it into a high-dimensional vector space. The model builds a prediction about the emotional state of the person who generated the sample by integrating the most important components and patterns in the input sequence using self-attentional processes. The proposed approach possesses a distinct advantage in its enhanced capacity to capture long-range correlations compared to conventional recurrent neural networks (RNN). This characteristic makes it particularly well-suited for the precise identification of emotions from samples of handwriting and drawings, signifying a notable advancement in the field of emotion detection. The proposed method produced cutting-edge outcomes of 92.64% on the benchmark dataset known as EMOTHAW (Emotion Recognition *via* Handwriting and Drawing).

# INTRODUCTION

Emotion detection (ED) is the process of recognizing and evaluating the emotional states and feelings of individuals using a variety of techniques. Accurately understanding and interpreting human emotions is the ultimate objective of ED, which has a variety of applications in areas including mental health, user experience, education, marketing, and security (*Acheampong, Wenyu & Nunoo-Mensah, 2020*). Emotions are one's reactions, and they can differ widely among individuals. Defining universal patterns or guidelines for detection might be challenging since individuals may show the same emotion in various ways. Although emotion science has made considerable strides, there is still much to learn about the subtleties and complexity of human emotions. It is still difficult to create comprehensive representations that adequately depict the entire spectrum of emotions (*Zad et al., 2021*). The development of intelligent systems to assist physicians at the point of treatment uses machine learning (ML) techniques. They can support conventional clinical examinations for the assessment of Parkinson's disease (PD) by detecting its early symptoms and signs. In patients with PD, previously taught motor abilities, including handwriting, are frequently impaired. This makes handwriting a potent identifier for the creation of automated diagnostic systems (*Impedovo, Pirlo & Vessio, 2018*). DL models have shown promising outcomes in ED, notably those built on RNNs and convolutional neural networks (CNNs). These models can recognize temporal connections and learn complicated patterns from the emotional data (*Pranav et al., 2020*).

The study conducted by *Kedar et al. (2015)* analyzed handwriting features such as baseline, slant, pen pressure, dimensions, margin, and boundary to estimate an individual's emotional levels. The study concludes that it will aid in identifying those individuals who are emotionally disturbed or sad and require psychiatric assistance to deal with such unpleasant emotions. Additionally, *Gupta et al. (2019)* examines electroencephalogram (EEG) signals from the user's brain to determine their emotional state. The study uses a correlation-finding approach for text improvement to change the words that match the observed emotion. The verification of the sentence's accuracy was then carried out utilizing a language modeling framework built on long short-term memory (LSTM) networks. In a dataset with 25 subjects, an accuracy of 74.95% was found when classifying five emotional states employing EEG signals. Based on handwriting kinetics and quantified EEG analysis, a computerized non-invasive, and speedy detection technique for mild cognitive impairment (MCI) was proposed in *Chai et al. (2023)*. They employed a classification model built on a dual-feature fusion created for medical decision-making. They used SVM with RBF kernel as the basic classifier and achieved a high classification rate of 96.3% for the aggregated features.

Existing ED research has mostly concentrated on a small number of fundamental emotions, such as happiness, sadness, and anger. The complexity and variety of emotional expressions make it difficult to adequately depict the entire emotional range. The performance of ED systems is improved by identifying significant characteristics across several modalities and creating suitable representations (*Zakraoui et al., 2023*). The capacity to recognize emotions *via* routine activities like writing and drawing could

contribute to mental health (*Rahman & Halim, 2023*). Collecting handwriting and drawing samples has been easier with the rise of human-machine interfaces like tablets (*Likforman-Sulem et al., 2017*). Therefore, a key component of the connection between human and computer communication is extracting and comprehending emotion (*Yang & Qin, 2021*).

In contrast to previous research, which mostly relied on text or audio data, this study involves a novel approach to analyze samples of handwriting and drawings to identify emotions. The suggested method employs a digital platform to analyze handwriting or drawing samples of an individual and forecast their emotional state using an attention-based transformer model. This method can capture more complex and sensitive feelings, which may not be capable of adequately conveyed through spoken words. This work employs an attention-based transformer model, which has proven to be very successful in natural language processing (NLP) tasks. The proposed model can capture long-range relationships and identify significant aspects in the input data, which may be essential in precisely recognizing emotions. The performance of the suggested model is tested on benchmark datasets that outperform the closely related work. The proposed study advances current information by establishing a baseline for evaluating the performance of ED models using handwriting and drawing examples.

## Challenges and limitations

It is interesting to improve communication between humans and robots by extracting emotions from handwriting. However, the amount and quality of the training dataset affect the process of creating predictive ML frameworks. There is not sufficient public data available to create precise models for emotion recognition from handwriting and drawing samples. The limited availability of public data poses challenges in constructing accurate models for emotion recognition from handwriting and drawing samples. Additionally, the absence of voice modality and facial expressions further complicates emotion detection in text (*Chatterjee et al., 2019*). Defining clear and unbiased criteria for different emotional states becomes challenging because individual handwriting styles can vary widely. Emotions are inherently complex, contributing to an overlap between distinct emotional states. For instance, it might be challenging to distinguish between sadness and frustration when they are conveyed in writing or drawing in similar ways. Emotions are frequently context-dependent and impacted by a variety of elements, such as personality, social environment, and cultural background. This makes it challenging to create models that correctly identify emotions in various contexts and circumstances. Despite these difficulties, researchers have been investigating numerous strategies to examine handwriting and determine emotional states. The accuracy of emotion recognition from handwriting can be increased by integrating many modalities, such as examining the content, structure, and dynamics of handwriting with other indications.

## Novelty and contribution

This research introduces several distinctive contributions to the proposed approach for emotion recognition from handwriting and drawing samples utilizing an attention-based

transformer model. Firstly, it integrates two modalities, handwriting and drawing, a departure from previous studies that often focused on one modality. This integration allows for a more comprehensive analysis of emotions, considering the diverse ways individuals express their feelings through these different mediums. The core contribution lies in the introduction of an attention-based transformer model, finely tuned to capture spatial and temporal correlations inherent in handwriting and drawing samples. Notably, the model excels in capturing long-term dependencies, a feature advantageous in understanding nuanced emotional expressions. Unlike traditional models, the transformer is not order-dependent, enhancing its flexibility in handling non-sequential input sequences such as handwriting and drawing samples. These contributions collectively advance the field of emotion recognition by offering a novel and effective model that considers diverse modalities and addresses inherent challenges in capturing emotional nuances.

## A SURVEY OF EXISTING WORK

In recent years, a substantial body of research has been dedicated to exploring the intersection of machine learning and computer vision in the domain of emotion detection (*Kanwal, Asghar & Ali, 2022*; *Azam, Ahmad & Haq, 2021*; *Asghar et al., 2022*; *Li & Li, 2023*). This has become particularly crucial given the growing significance of understanding and interpreting human emotions. Beyond emotion detection, machine learning has found applications in various domains, including hot topic detection (*Khan et al., 2017*, *2023*), anomaly detection (*Haq & Lee, 2023*), and recognizing anomalous patterns (*Haq et al., 2023*), especially in social media contexts. Furthermore, machine learning has played a pivotal role in text classification (*Ullah et al., 2023*; *Khan et al., 2022*), providing sophisticated approaches for organizing and categorizing textual data and rating software applications. The ability to classify information is vital not only for sentiment analysis but also for broader applications such as content categorization (*Zhang et al., 2023*). In addition, information fusion (*Wang et al., 2021*; *Zhang et al., 2020*), where data from diverse sources are integrated to provide a more comprehensive understanding, has gained prominence. In the pursuit of accurate diagnoses for depression, *Esposito et al. (2020)* have explored innovative methodologies to obtain more reliable measurements than conventional questionnaires. This involves analyzing behavioral data, including voice, language, and visual cues, with sentiment analysis techniques evolving from linguistic characteristics to sophisticated tools for text, audio, and video recordings. The research study (*Aarts, Jiang & Chen, 2020*) introduces an application detecting four distinct emotions from social media posts, outlining techniques, outcomes, challenges, and proposed solutions for the project's continuity. The research study (*Aarts, Jiang & Chen, 2020*) utilizes convolutional neural networks to assess visual attributes in characterizing graphomotor samples of Parkinson's disease (PD) patients. The study achieves an 83% accuracy in early PD prediction using a dataset of 72 subjects through visual information and SVM classification. According to the study (*Aouraghe et al., 2020*) emotions like stress, worry, and depression have an impact on health, so it is crucial to recognize their emotional symptoms as early as possible. The study conducted in *Moetesum et al. (2019)*

focused on the possibility of using handwriting's visual characteristics to anticipate PD. The structure and operations of some brain areas are impacted by neurodegenerative illnesses, which lead to a gradual deterioration in cognitive, functional, and behavioral abilities.

A research study (*Ayzeren, Erbilek & Çelebi, 2019*) introduces a distinctive handwriting and signature biometric database with emotional status labels. The investigation delves into predicting emotional states (happy, sad, stress) from online biometric samples, achieving noteworthy success, especially in stress prediction from handwriting. The database, encompassing 134 participants, includes demographic information for comprehensive analysis. Examining preserved graphic structures in handwriting, the study (*Rahman & Halim, 2022*) explores the correlation between handwriting features and personality traits. Using a graph-based approach, eleven features are extracted to predict personality traits based on the Big Five model. Employing a semi-supervised generative adversarial network (SGAN) for enhanced accuracy, the study achieved a remarkable predictive accuracy of 91.3%. *Nolazco-Flores et al. (2021)* characterizes emotional states related to depression, anxiety, and stress using features from signals captured on a tablet. The EMOTHAW database includes handwriting and drawing tasks categorized into specific emotional states. Selected features improve average accuracy classification (up to 15%) compared to the baseline. The study conducted in *Shrivastava, Kumar & Jain (2019)* presented a DL framework for the challenge of fine-grained emotion identification utilizing multimodal text data. They unveiled a new *corpus* that depicts various emotions gleaned from a TV show's transcript. They employed a sequence-based CNN with word embedding as an ED framework to identify emotions. The study conducted in *Bhattacharya, Islam & Shahnawaz (2022)* provided a unique approach based on the Agglomerative Hierarchical Clustering algorithm, which can recognize the emotional state of the individual by examining the image of the handwritten text. They identify emotions with a maximum accuracy of 75%. There exists a lack of well-established techniques for emotion recognition using handwriting and drawing samples. The precise characteristics or patterns in handwriting that relate to certain emotions are not well discovered. Various experiments are conducted to detect emotion from handwritten text, however, there still exist certain limitations which are explained in the following sub-section.

## PROPOSED METHODOLOGY

The proposed study introduces an attention-based transformer model designed to generate a more comprehensive feature map from handwriting and drawing samples. This model aims to accurately identify both handwritten information and emotional content. The model supposes that writing and drawing are impacted by one's emotional state and are connected to behavior. The proposed method involves gathering an individual's writing through an electronic device and analyzing it to determine her emotional state. The suggested model is founded on the transformer architecture, which makes use of attention processes to provide a more detailed feature map of the data. The goal is to enhance the model's capacity to recognize and understand the information and feelings represented in handwriting and drawing. The suggested study offers a thorough assessment of numerous

writing and drawing traits and identifies their relationship to emotional states. Using writing and drawing samples, this work identified emotions with a high degree of accuracy. The details of how the algorithm works are explained in Algorithm 1.

Utilizing the Attention-Based Transformer Model for Emotion Detection involves a systematic process. First, the required libraries and frameworks are imported, and the model architecture is defined, considering parameters such as the number of layers (L), dimension of the model (d_model), number of attention heads (h), and dimension of the feed-forward network (dff). Pre-trained weights of the Transformer model are loaded, and the provided handwriting and drawing samples undergo preprocessing, converting them into a suitable input format and storing them for subsequent use. Following this, the preprocessed samples are loaded, converted into tensors, and passed through the Transformer model for inference. The model's output is processed through a linear layer and a softmax function to generate predicted probabilities for each emotion. The final step involves extracting the emotion with the highest probability for each sample, ultimately returning the predicted emotion. This streamlined process ensures effective and nuanced emotion detection from input samples.

## Emotion models

The core of ED systems is emotional models used to represent individual feelings. It is crucial to establish the model of emotion to be used before beginning any ED-related activity.

### Discrete emotion models

The discrete model of emotions includes categorizing emotions into several groups or subgroups. The Paul Ekman model (*Ekman, 1999*) classifies emotions into six fundamental categories. According to the idea, there are six essential emotions, which are independent of one another and come from various neurological systems depending on how an individual experiences a scenario. These basic feelings include happiness, sadness, anger, disgust, surprise, and fear. However, the synthesis of these feelings may result in more complicated feelings like pride, desire, regret, shame, and so on.

### Dimensional emotion models

The dimensional model requires that emotions be placed in a spatial location because it assumes that emotions are not independent of one another and that there is a relationship between them. The circumplex of affect is a circular, two-dimensional concept presented by *Russell (1980)* that is significant in dimensional emotion expression. The Arousal and Valence domains of the model separate emotions, with Arousal classifying emotions according to activations and deactivations and Valence classifying emotions according to pleasantness and unpleasantness. This work uses both emotional models.

### Experimental design

The implementation of the proposed work is done using Jupyter Notebook with a five-fold cross-validation training strategy. Both the EMOTHAW and SemEval datasets were used

---

**Algorithm 1 Emotion detection using attention-based transformer model.**

Input: Handwriting and drawing samples

Output: Predicted emotion

1. Import required libraries and frameworks

2. Define the Transformer model architecture:

    - Let L represents the number of layers in the Transformer model.

    - Let d_model represent the dimension of the model.

    - Let h represent the number of attention heads.

    - Let dff represents the dimension of the feed-forward network.

3. Load pre-trained weights of the Transformer model.

4. Preprocess samples:

    - Let N be the number of samples.

    - For i = 1 to N:

        a. Apply preprocessing steps to the i-th sample.

        b. Convert the preprocessed i-th sample into a suitable input format.

        c. Store the preprocessed i-th sample.

5. Load the preprocessed samples.

6. Convert the preprocessed samples into tensors.

7. Perform inference using the loaded model:

    - Let M represents the maximum sequence length.

    - Let V represents the vocabulary size.

    - Let S represents the number of emotions.

    - For i = 1 to N:

        a. Pass the preprocessed i-th sample through the Transformer model:

            - Encoder output: enc_output = Encoder(input)

            - Decoder output: dec_output = Decoder (target, enc_output)

        b. Apply a linear layer to the decoder output: logits = Linear(dec_output)

        c. Apply a softmax function to obtain the predicted probabilities: probabilities = Softmax(logits)

        d. Store the predicted probabilities for the i-th sample.

8. Extract the predicted emotions:

    - For i = 1 to N:

        a. Extract the emotion with the highest probability from the i-th sample.

9. Return the predicted emotion.

End

---

during training. The learning rate is set to 0.0001, and the model is trained over 25 epochs. A weight decay of 0.05 is applied to control overfitting, and the Adam optimizer is employed for efficient parameter updates. The loss function utilized for training is cross

entropy. Additionally, attention heads are incorporated using a multi-heads mechanism, enhancing the model's ability to capture intricate patterns and dependencies within the data. The model's number of layers, attention heads, the dimensionality of embeddings, learning rate, and batch size are among the hyperparameters that are fine-tuned. Several assessment criteria, including accuracy, F1 score, precision, and recall, are used to assess the model's performance on the test set. The number of samples for each emotion category that were successfully and wrongly identified is used to calculate these measures.

## Datasets

### EMOTHAW dataset

The EMOTHAW database (*Likforman-Sulem et al., 2017*) contains samples from 129 individuals (aged between 21–32) whose emotional states, including anxiety, depression, and stress, were measured using the Depression Anxiety Stress Scales (DASS) assessment. Due to the dearth of publicly accessible labeled data in this field, this database itself is a helpful resource. A total of 58 men and 71 women participated in the dataset. The age range has been constrained to decrease the experiment's inter-subject variation. Seven activities are recorded using a digitizing tablet: drawing pentagons and houses, handwriting words, drawing circles and clocks, and copying a phrase in cursive writing. The writing and drawing activities used to get the measures are well-researched exercises that are also used to assess a person's handwriting and drawing abilities. Records include pen azimuth, altitude, pressure, time stamp, and positions (both on article and in the air). The generated files have the svc file extension, generated through the Wacom device. Figure 1 shows a system overview of the proposed method.

### SemEval dataset

The Semantic Evaluations (SemEval) dataset (*Rosenthal, Farra & Nakov, 2019*) includes news headlines in Arabic and English that were taken from reputable news sources including the BBC, CNN, Google News, and other top newspapers. There are 1,250 total data points in the dataset. The database contains a wealth of emotional information that may be used to extract emotions, and the data is labeled according to the six emotional categories proposed by *Ekman (1999)* (happiness, sadness, fear, surprise, anger, and disgust).

## Feature extraction

Drawing and handwriting signals are examples of time series data, which is displayed as a collection of data points gathered over time. In this study, we extract characteristics from the handwritten and drawing signals in the time domain, frequency domain, and statistical domain. The signal's changing amplitude over time is used to extract time-domain characteristics, which contain the signal's mean and standard deviation. The frequency content of the signal is extracted to get frequency-domain characteristics, in which spectral entropy, spectral density, and spectral centroid are included. The signal's statistical characteristics are used to extract statistical features. Examples consist of mutual information, cross-correlation, and auto-correlation.
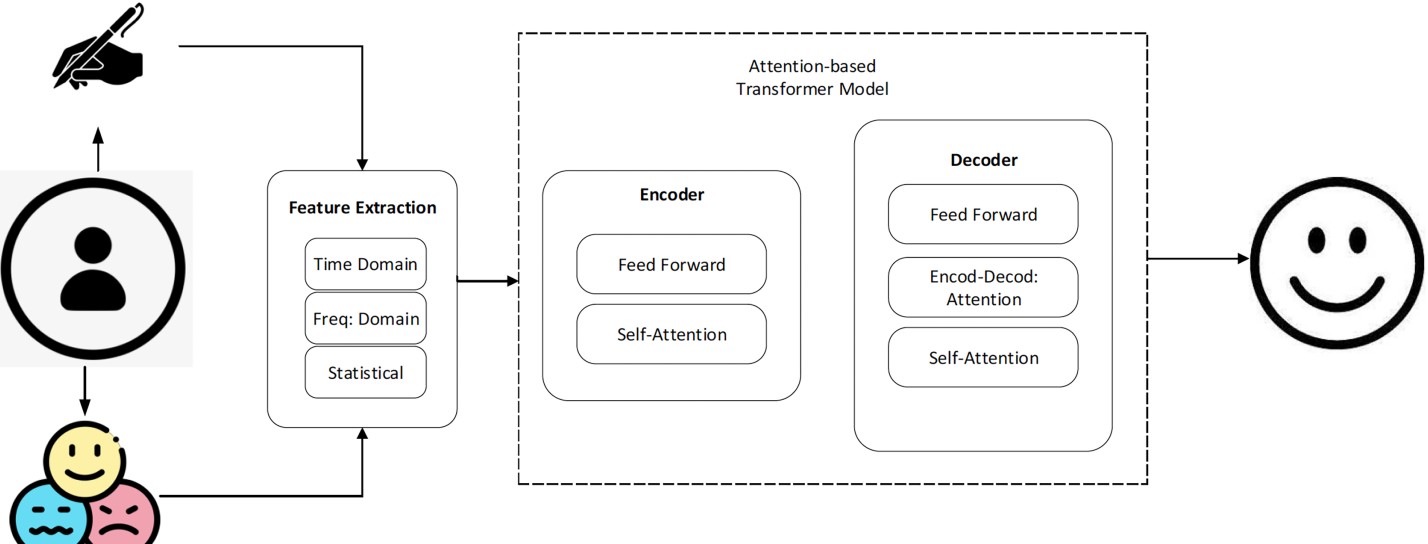

**Figure 1 The basic steps of the proposed work.**

### *Mean*

The average value of the signal over a particular time interval is calculated to get an idea about the overall level of the signal. In this work, the mean value of a handwriting signal $x[n]$ over a time interval of $N$ samples is calculated as:

$$\mu = (1/N) * \Sigma\, x[n] \tag{1}$$

where $\mu$ is the mean value of the signal, $n$ ranges from $0$ to $N-1$, and $\Sigma$ denotes the sum of all values.

### *Standard deviation*

The standard deviation of the signal over a particular time interval is calculated to measure the amount of variability in the signal. In this work, the standard deviation of a handwriting signal $x[n]$ over a time interval of $N$ samples are calculated as:

$$\sigma = \sqrt{(1/N)\sum (x[n]-\mu)^2} \tag{2}$$

where $\sigma$ represents the standard deviation of the signal, $n$ ranges from $0$ to $N-1$.

### *Spectral density*

Spectral density is used to measure the power distribution of a handwriting signal in the frequency domain. In this work, the spectral density of a handwriting signal is calculated as:

$$S(f) = |F(f)|^2 \tag{3}$$

where $S(f)$ represents the spectral density, and $F(f)$ denotes the Fourier transform of the signal.

### Spectral entropy

In the proposed work, spectral entropy is used to measure the irregularity of the power spectrum in the handwriting signal for feature extraction. The mathematical formula to calculate spectral entropy is given by:

$$SE = -\sum_{i=1}^{n} P_i * \log (P_i) \tag{4}$$

where $SE$ represents the spectral entropy, $P_i$ shows the power at the $i - th$ bin of the power spectrum, and $N$ denotes the number of frequency bins.

### Autocorrelation

Autocorrelation measures the degree of similarity between the handwriting signal and a delayed version of itself. It can be used to identify patterns or repeating features in the signal. In this work, the autocorrelation of a handwriting signal $x[n]$ over a time interval of $N$ samples are calculated as:

$$R[k] = (1/N) * \Sigma\, x[n] * x[n - k] \tag{5}$$

where $R[k]$ is the autocorrelation of the signal at lag $k$, $n$ ranges from 0 to $N - 1$.

## Classification

An attention-based transformer model is used to classify the features of handwriting and drawing signals. The pre-processed data is delivered into the attention-based transformer model during training to decrease the discrepancy between the model's forecasts and the actual labels assigned to the data. In this study, the model is trained by employing strategies like gradient descent and backpropagation. The capacity to pay attention to various aspects of the incoming data is one of the main characteristics of an attention-based transformer model. This is accomplished through the use of an attention mechanism, which enables the model to concentrate on the input's key characteristics at each stage of the classification process. The model can acquire the ability to recognize patterns and traits that are crucial for differentiating between various classes of handwriting and drawing by paying attention to different components of the input. Additionally, this architecture often has several processing levels. Every layer of the model is made to learn more intricate representations of the input data, enabling the model to identify subtler characteristics and patterns. Typically, a softmax function is used to generate a probability distribution across all feasible classes of handwriting and drawing using the output of the last layer.

## EXPERIMENTS

The initial stage in the experiments is to process the data from each dataset to identify the appropriate features for emotion identification from handwriting and drawing examples. An attention-based transformer model for emotion recognition from handwriting and drawing samples is trained and evaluated through a series of experiments. For this experiment, we use the EMOTHAW and SemEval benchmark datasets. These datasets include several types of handwritten and drawn samples as well as the associated emotion

labels. An SVC file containing 1,796 points with seven metrics (x location, y location, time stamp, pen status, azimuth, altitude, and pressure) is part of the EMOTHAW dataset. Both points acquired on article (pen status equal to 1) and points acquired in the air (pen status equal to 0) are included. The SemEval dataset includes 4,359 handwritten text samples of various emotions, such as happiness, surprise, joy, anger, fear, and sadness.

## Attention-based transformer model

The capacity of the attention-based transformer model to identify long-range relationships in sequential data has made it a popular neural network design in recent years. The model is made up of several self-attention layers that enable it to intelligently focus on various input data components depending on their applicability to the work at hand. The transformer architecture's attention methods are used to direct the model's attention to the areas of the input data that are most important. The performance of the final classification is improved by attention methods, which collect additional factors that are helpful for classification and give higher priority to essential elements of information. The attention-based transformer model is employed in the classification of handwriting signals to develop an illustration of the input signal that captures the required characteristics for inferring the associated emotion. The model produces a probability distribution across the various emotion categories after receiving as input a series of time-domain, frequency-domain, or statistical data derived from the handwritten signal.

Input encoding: In the proposed work, the input sequence of extracted features undergoes an initial encoding through a linear transformation and a subsequent non-linear activation function to generate a sequence of embeddings.

$$X = \{x_1, x_2, \ldots, x_n\} \rightarrow E = \{e_1, e_2, \ldots, e_n\},$$

$$\text{where } E_i = f\left(\sum_i W_i x_i + b\right). \tag{6}$$

Self-Attention: The proposed work utilizes a self-attention layer to compute a weighted sum of the embeddings, with the weights determined based on the similarity between each pair of embeddings.

$$Z = self\_attention(E) = softmax\left(\frac{Q * K^T}{\sqrt{d_k}}\right) * V \tag{7}$$

where $Q$, $K$, and $V$ represent the query, key, and value matrices, respectively, and $d_k$ denotes the dimensionality of the key vectors. The softmax function is used to normalize the similarity scores to obtain a probability distribution over the keys. In a broader sense, the attention mechanism is conceptually explained by the hypothetical formula: Attention $q, k, v$. This involves assessing the similarity (Sim) between queries ($q$) and keys ($k$), followed by multiplying this similarity score with the weighted sum of values ($v$). Put simply, it determines the level of attention each query should assign to its corresponding key and combines these attention-weighted values to produce the final output. Typically,

Softmax is employed to derive the similarity score. The Softmax-based calculation of the attention mechanism can be expressed by the equation:

$$attention(q, k, v) = softmax\left(\frac{q * k^t}{\sqrt{d_k}}\right). \tag{8}$$

The Softmax attention mechanism entails matrix multiplication, where the dot product is computed between each feature vector in q and the transpose of k. This dot product is then divided by the scaling factor $\sqrt{d_k}$ before being subjected to a softmax function. The Softmax attention mechanism relies on the dot product operation, which considers both the angle and magnitude of vectors when computing similarity.

Multi-head attention: To capture multiple aspects of the input writing signal in this work, the self-attention layer is extended to include multiple heads.

$$Z_i = self\_attention_i(E) = softmax\left(\frac{Q_i * K_i^T}{\sqrt{d_k}}\right) * V_i \tag{9}$$

where $i$ represents the head index, and $Q_i$, $K_i$, and $V_i$ denote the query, key, and value matrices for the $i$-th head.

Feed-forward networks: In this work, the output of the self-attention layer is passed through a feed-forward network to further refine the representation.

$$Y = f(feedforward(Z)) \tag{10}$$

where $f$ represent a non-linear activation function. Here the feed-forward network consists of two linear transformations with a non-linear activation function.

Output prediction: In this work, the final output of the transformer model is obtained by passing the refined representation through a linear transformation and a Softmax function.

$$P = softmax(W_i * Y + b) \tag{11}$$

where $P$ represents the predicted probability distribution over the emotion categories.

## Evaluations metrics

The assessment criteria employed in emotion identification from handwriting and drawing samples employing an attention-based transformer model largely rely on the particular task and dataset being used. The following is an explanation of the assessment metrics utilized in this study.

### Accuracy

This measures the proportion of correctly classified emotions to the total number of emotions used in the dataset.

$$Accuracy = \left(T_p + T_N\right) / \left(T_p + T_N + F_p + F_N\right) \tag{12}$$

where $T_p$ represents the number of true positive samples (samples that were correctly classified as the target emotion), $T_N$ denotes the number of true negative samples (samples

that were correctly classified as not the target emotion), $F_p$ shows the number of false positive samples (samples that were incorrectly classified as the target emotion), and $F_N$ describes the number of false negative samples (samples that were incorrectly classified as not the target emotion).

### F1 score

The F1 score, which offers a fair evaluation of the model's achievement, is a harmonic mean of accuracy and recall. In contrast to recall, which assesses the proportion of genuine positive predictions made from all positive samples, precision assesses the proportion of true positive predictions made from all positive predictions.

$$Precision = \frac{T_p}{T_p + F_p} \tag{13}$$

$$Recall = \frac{T_p}{T_p + F_N} \tag{14}$$

$$F_1 = 2 * \left(\frac{Precision * Recall}{Precision + Recall}\right). \tag{15}$$

$F_1$ represents a weighted average of precision and recall, with a maximum value of 1 and a minimum value of 0.

## RESULTS AND DISCUSSION

In the first experiment harnessing the rich EMOTHAW dataset, the proposed study employed an attention-based transformer model to meticulously analyze collected features and unravel the intricacies of emotional states. The model demonstrated its prowess on the test set, surpassing conventional machine learning techniques with a peak accuracy of 92.64%. Notably, the integration of both handwriting and drawing traits, as outlined in Table 1, proved to be a game-changer, giving superior accuracy in diagnosing depression and offering nuanced insights into the identification of anxiety and stress.

Using the EMOTHAW dataset, the proposed study used an attention-based transformer model for the collected features to achieve the recognition results for the three emotional states. On the test set, the model outperformed conventional machine learning techniques with the highest accuracy of 92.64%. Using both handwriting and drawing traits together, as shown in Table 1, we were able to diagnose depression with the best accuracy. Similar findings were obtained for the identification of anxiety, with 79.51% accuracy using drawing, 77.38% using writing, and 83.22% integrating both writing and drawing features. For stress detection, we obtained the highest accuracy of 79.41% using writing features. To encapsulate, the EMOTHAW dataset, coupled with the proposed advanced model, not only advances the field of emotion detection but also sets a new standard in accuracy, particularly when considering the synergistic effects of both handwriting and drawing traits. These findings underscore the robustness of the proposed approach and its potential impact on applications ranging from mental health diagnostics to personalized well-being assessments.

**Table 1 Results obtained using the EMOTHAW dataset.**

| Emotion | Task | Accuracy (%) |
|---|---|---|
| Depression | Drawing | 86.15 |
| | Writing | 91.39 |
| | Both | 92.64 |
| Anxiety | Drawing | 79.51 |
| | Writing | 77.38 |
| | Both | 83.22 |
| Stress | Drawing | 78.76 |
| | Writing | 79.41 |
| | Both | 78.05 |

In the second experiment, incorporating the diverse SemEval dataset enriched with emotional annotations from tweets, blog posts, and news articles, the study examined text-based emotion detection. The categorization of features into three distinct emotional states, angry, happy, and sad, unfolded insightful revelations. Notably, the proposed model showcased state-of-the-art performance on the test dataset, particularly excelling in the identification of sadness with a remarkable F1 score of 87.06%, as clarified in Table 2. Moreover, the proposed study achieved notable success in detecting happy and angry states, reaffirming the versatility of the model across various emotional dimensions.

In the SemEval dataset, a variety of text samples with emotional annotations are included, such as tweets, blog posts, and news articles. In this work, the features are categorized into three different emotional states, angry, happy, and sad. On the test dataset, the model produced state-of-the-art results for sad state identification, with an F1 score of 87.06%, as shown in Table 2. For happy state detection, we obtained the highest F1 score of 79.73% and for angry state detection we obtained the highest F1 score of 83.12%. To sum up, the SemEval dataset, coupled with the proposed model, opens new horizons in text-based emotion detection. The robust performance across different emotional states, especially the outstanding identification of sadness, highlights the adaptability and efficacy of the proposed work. These outcomes not only contribute to the academic discourse but also pave the way for practical applications in sentiment analysis across diverse textual genres.

Figure 2 showcases the robust performance of the proposed model. This high level of accuracy indicates the model's proficiency in learning from the training data and effectively generalizing it to new, unseen data during testing. The sustained elevation of both lines above 90% emphasizes the reliability and effectiveness of the proposed model in accurately predicting emotional states based on the provided features. To specifically address color differentiation issues, we have manually introduced circles to represent training accuracy and rectangles for testing accuracy. This visual distinction aims to enhance clarity and inclusivity for a diverse audience, mitigating potential challenges associated with color perception.

**Table 2 Results obtained using SemEval dataset.**

| Emotion | Metrics | Score (%) |
|---|---|---|
| Angry | Precision | 80.13 |
| | Recall | 86.35 |
| | F1 | 83.12 |
| Happy | Precision | 83.51 |
| | Recall | 76.28 |
| | F1 | 79.73 |
| Sad | Precision | 88.65 |
| | Recall | 85.54 |
| | F1 | 87.06 |

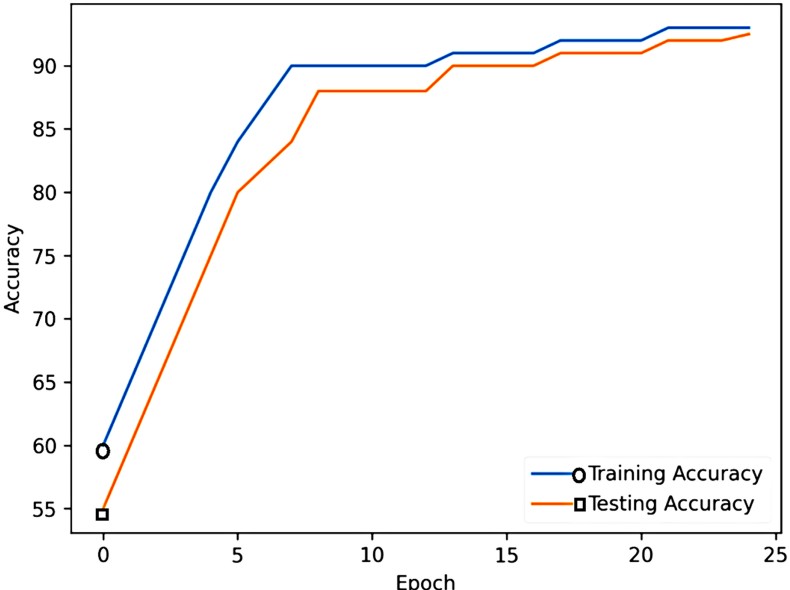

**Figure 2 Training and testing accuracy of the proposed model.**

## Results comparison

The section conducts a comprehensive analysis of the results, drawing comparisons with existing datasets and state-of-the-art approaches. Table 3 provides a detailed examination of the model's performance using the EMOTHAW dataset, highlighting its accuracy in emotion state recognition. Moving forward, Table 4 extends the comparison to the SemEval dataset, offering insights into how the proposed model fares in diverse text samples like tweets, blog posts, and news articles. Additionally, Table 5 positions the obtained results in the broader context of state-of-the-art methodologies, underlining the competitiveness and advancements achieved by the proposed study.

Table 3 shows the comparison of the results using the EMOTHAW dataset. The study conducted by *Likforman-Sulem et al. (2017)* used a random forest (RF) technique to

**Table 3 Results comparison using the EMOTHAW dataset.**

| Study | Year | Emotion | Task | Accuracy (%) |
|---|---|---|---|---|
| *Likforman-Sulem et al. (2017)* | 2017 | Depression | Drawing | 72.80 |
| | | | Writing | 67.80 |
| | | | Both | 71.20 |
| | | Anxiety | Drawing | 60.50 |
| | | | Writing | 56.30 |
| | | | Both | 60.00 |
| | | Stress | Drawing | 60.10 |
| | | | Writing | 51.20 |
| | | | Both | 60.20 |
| *Nolazco-Flores et al. (2021)* | 2021 | Depression | Drawing | 75.59 |
| | | | Writing | 80.31 |
| | | | Both | 74.01 |
| | | Anxiety | Drawing | 67.71 |
| | | | Writing | 68.50 |
| | | | Both | 72.44 |
| | | Stress | Drawing | 67.71 |
| | | | Writing | 67.71 |
| | | | Both | 70.07 |
| *Rahman & Halim (2023)* | 2023 | Depression | Drawing | 83.28 |
| | | | Writing | 89.21 |
| | | | Both | 87.11 |
| | | Anxiety | Drawing | 76.12 |
| | | | Writing | 74.54 |
| | | | Both | 80.03 |
| | | Stress | Drawing | 75.39 |
| | | | Writing | 75.17 |
| | | | Both | 74.38 |
| Proposed work | 2023 | Depression | Drawing | 86.15 |
| | | | Writing | 91.39 |
| | | | Both | 92.64 |
| | | Anxiety | Drawing | 79.51 |
| | | | Writing | 77.38 |
| | | | Both | 83.22 |
| | | Stress | Drawing | 78.76 |
| | | | Writing | 79.41 |
| | | | Both | 78.05 |

analyze and categorize the collection of characteristics collected from the EMOTHAW dataset, which is a machine learning algorithm that uses a group of decision trees and a feature ranking mechanism. They achieved a higher accuracy of 72.8% for depression detection using drawing features. Similarly, they achieved a higher accuracy of 60.50% for

**Table 4 Results comparison on SemEval dataset.**

| Study | Emotion | Metrics | Score (%) |
|---|---|---|---|
| *Chatterjee et al. (2019)* | Angry | Precision | 77.23 |
| | | Recall | 84.23 |
| | | F1 | 80.58 |
| | Happy | Precision | 80.40 |
| | | Recall | 70.77 |
| | | F1 | 75.28 |
| | Sad | Precision | 84.94 |
| | | Recall | 81.20 |
| | | F1 | 83.03 |
| Proposed work | Angry | Precision | 80.13 |
| | | Recall | 86.35 |
| | | F1 | 83.12 |
| | Happy | Precision | 83.51 |
| | | Recall | 76.28 |
| | | F1 | 79.73 |
| | Sad | Precision | 88.65 |
| | | Recall | 85.54 |
| | | F1 | 87.06 |

**Table 5 Comparison with state-of-the-art approaches.**

| Study | Year | Model | Dataset | Accuracy |
|---|---|---|---|---|
| *Rahman & Halim (2023)* | 2023 | Combined features with BiLSTM | EMOTHAW | 89.21% |
| *Nolazco-Flores et al. (2021)* | 2021 | Combined features with SVM | EMOTHAW | 80.31% |
| *Gavrilescu & Vizireanu (2018)* | 2018 | Artificial neural networks (ANN) | 128 subjects | 84.4% |
| *Mostafa, Al-Qurishi & Mathkour (2019)* | 2019 | Decision Tree, *k*-NN | 83 subjects | 68.67% |
| *Chitlangia & Malathi (2019)* | 2019 | Support vector machine (SVM) | 50 subjects | 80.0% |
| **Proposed work** | **2023** | **Combined features with attention-based transformer** | **EMOTHAW** | **92.64%** |

anxiety detection and 60.20% for stress detection. The work conducted by *Rahman & Halim (2023)* used a combination of temporal, spectral, and Mel Frequency Cepstral Coefficient (MFCC) approaches to extract characteristics from each signal and discover a link between the signal and the emotional states of stress, anxiety, and sadness. They classified the vectors of the generated characteristics using a Bidirectional Long-Short Term Memory (BiLSTM) network. They obtained a higher accuracy of 89.21% for depression detection using writing features. For anxiety detection, they achieved a higher accuracy of 80.03% using both drawing and writing features. For stress detection, they achieved a higher accuracy of 75.39% using drawing features. The study conducted in *Nolazco-Flores et al. (2021)* used the fast correlation-based filtering approach to choose the optimal characteristics. The retrieved features were then supplemented by introducing a

tiny amount of random Gaussian noise and a proportion of the training data that was randomly chosen. A radial basis SVM model is trained and obtained the higher accuracy of 80.31% for depression detection. However, the proposed work achieved the highest accuracy of 92.64% for depression detection using both drawing and writing features. Similarly, for anxiety detection, we achieved the highest accuracy of 83.22% using both drawing and writing features. For stress detection, we achieved the highest accuracy of 79.41% using writing features.

In contrast to prior studies, particularly the work by *Esposito et al. (2020)* employing emotion detection from text through the BiLSTM network, the proposed work, as shown in Table 4, exhibits noteworthy achievements using the SemEval dataset. While *Esposito et al. (2020)* achieved a commendable F1 score of 83.03% for sad state detection, the proposed study surpassed these results. We attained the highest F1 score of 83.12% for angry state detection, 79.73% for happy state detection, and an exceptional 87.06% for sad state detection. This underscores the efficacy and robustness of our proposed approach, establishing its superiority across diverse emotional states.

In the realm of emotion detection, various state-of-the-art approaches have been explored, as exemplified by studies such as *Rahman & Halim (2023)*, *Nolazco-Flores et al. (2021)*, *Gavrilescu & Vizireanu (2018)*, *Mostafa, Al-Qurishi & Mathkour (2019)* and *Chitlangia & Malathi (2019)* in Table 5. These endeavors utilized different methodologies and sample sizes to decipher emotional states. Specifically, *Gavrilescu & Vizireanu (2018)*, *Mostafa, Al-Qurishi & Mathkour (2019)*, and *Chitlangia & Malathi (2019)* employed artificial neural networks (ANN), decision trees, and k-NN with 128, 83, and 50 subjects, respectively. Meanwhile, *Rahman & Halim (2023)* employed combined features with BiLSTM, and *Nolazco-Flores et al. (2021)* leveraged combined features with a Support Vector Machine (SVM). The reported accuracy results from these studies ranged from 68.67% to 89.21%. Notably, the proposed model surpassed all these benchmarks, achieving an impressive 92.64% accuracy using the EMOTHAW dataset. This attests to the superior performance and efficacy of the proposed approach compared to existing state-of-the-art methods in the field.

## Threats to analysis

The proposed approach for emotion detection from handwriting and drawing samples exhibits promising results, although certain inherent limitations merit consideration. Firstly, the model's performance is contingent on the quality and representativeness of the training dataset, with potential biases affecting generalizability. Additionally, the limited diversity in handwriting and drawing styles within the training data may impact the model's adaptability to extreme variations in individual expression. Cultural nuances in emotional expression pose another challenge, as the model's performance may vary across diverse cultural contexts. Dependency on machine translation tools for languages beyond the training set introduces potential errors, and the predefined set of emotions in focus might not capture the full spectrum of human emotional expression. Ethical considerations regarding privacy and consent in deploying emotion detection technologies add another layer of complexity. While these limitations are acknowledged, the proposed

model serves as a foundational step, emphasizing the need for ongoing research and refinement to address these challenges and enhance overall robustness in varied contexts.

## Discussion

Text-based ED is focused on the feelings that lead people to write down particular words at specific moments. According to the results, multimodal ED, such as voice, body language, facial expressions, and other areas, receive more attention than their text-based counterparts. The dearth has mostly been caused by the fact that, unlike multimodal approaches, texts may not exhibit distinctive indications of emotions, making the identification of emotions from texts significantly more challenging compared to other methods. Because there are no facial expressions or vocal modulations in handwritten text, this makes emotion detection a difficult challenge. The purpose of this work was to ascertain the level of interest in the field of handwritten text emotion recognition. To perform classification and analysis tasks, handwriting and drawing signals are processed using the feature extraction steps to isolate significant and informative attributes. In this study, we found that individual variances in handwriting characteristics were caused by their emotional moods. Incorporating more characteristics like pressure and speed into the input data, we saw that the attention-based transformer model obtained great accuracy. We observed that adding additional features can enhance the model's performance even more.

## CONCLUSION

The drawing and handwriting signals are instances of time series data, which is shown as a collection of data points acquired over time. In this work, we extract time-domain, frequency-domain, and statistical-domain features from the handwriting and drawing signals. The proposed model has the benefit of being able to capture long-range relationships in the input data, which is especially beneficial for handwriting and drawing samples that contain sequential and spatial information. The model's attention mechanism also enables it to concentrate on relevant components and structures in the input data, which may enhance its capacity to recognize minor emotional signals. The hyperparameters that are adjusted during the testing of the model include the number of layers, attention heads, the dimensionality of embeddings, learning rate, and batch size. Concerning accuracy and F1 scores, the attention-based transformer model used in this study excelled on two benchmark datasets.

In the future, transfer learning techniques could be used to pre-train the attention-based transformer model on large datasets and fine-tune it for specific emotion detection tasks, which could potentially improve the model's performance on smaller datasets.

### Funding

This research is funded by the Researchers Supporting Project Number (RSPD2024R947), King Saud University, Riyadh, Saudi Arabia. The funders had no role in study design, data collection and analysis, decision to publish, or preparation of the manuscript.

## Grant Disclosures

The following grant information was disclosed by the authors:
Researchers Supporting Project, King Saud University, Riyadh, Saudi Arabia:
RSPD2024R947.

## Competing Interests

Khursheed Aurangzeb is an Academic Editor for PeerJ.

## Author Contributions

- Zohaib Ahmad Khan conceived and designed the experiments, performed the experiments, performed the computation work, prepared figures and/or tables, and approved the final draft.
- Yuanqing Xia analyzed the data, performed the computation work, authored or reviewed drafts of the article, and approved the final draft.
- Khursheed Aurangzeb analyzed the data, performed the computation work, authored or reviewed drafts of the article, and approved the final draft.
- Fiza Khaliq conceived and designed the experiments, performed the computation work, authored or reviewed drafts of the article, and approved the final draft.
- Mahmood Alam performed the experiments, prepared figures and/or tables, and approved the final draft.
- Javed Ali Khan conceived and designed the experiments, authored or reviewed drafts of the article, and approved the final draft.
- Muhammad Shahid Anwar performed the experiments, prepared figures and/or tables, and approved the final draft.

## Data Availability

The code is available in the Supplemental File.

The datasets used in this study are available at:

- The "EMOTHAW" dataset: https://www.psicologia.unicampania.it/the-lab/our-activities (Laurence Likforman-Sulem, Anna Esposito, Marcos Faundez-Zanuy, Stephan Clemencon, Franceand Gennaro Cordasco).

- The "SemEval" dataset: https://alt.qcri.org/semeval2016/task4/index.php?id=data-and-tools (Sara Rosenthal, QatarNoura Farra, Preslav Nakov).

## Supplemental Information

Supplemental information for this article can be found online at http://dx.doi.org/10.7717/peerj-cs.1887#supplemental-information.

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
