# Peer review of "Emotion detection from handwriting and drawing samples using an attention-based transformer model"

_PeerJ Computer Science, doi:10.7717/peerj-cs.1887_

## Round 0.1 · original submission · Major Revisions

Please see the comments from reviewers and address them in a revision.

Reviewer 1 ·

Basic reporting

The article “Emotion Detection” by Khan et al., presents an emotion detection framework to address the gap left by previous work. The framework integrates the transformer model to address the issue. The article at hand is well-written, well-organized, and addresses an important real-world problem. However, the reviewer found some major issues while reviewing the article. The detailed comments are given below.

Experimental design

1. A section about experimental setup should be added to the paper, as well as the epochs and hyperparameters.
2. Provide more details about the use of the SoftMax function in the attention-based transformer model.
3. No diagram for an attention-based transformer is provided, better include it.

Validity of the findings

1. Since Emotion detection is an important topic, how about adding a “threat to the validity” paragraph before the discussion section?
2. The tables are well presented but no text to represent the results, more written explanations of the Tables and Figures will increase the worth of the paper.

Additional comments

1. The abstract of the paper doesn’t seem to have connectivity; sentence structure needs to be re-arranged/re-modified.
2. In the survey/related work section, the author needs to discuss related work from the field, not give definitions.
3. The paper does not include citations from related journal articles. Incorporating relevant citations can improve the paper's academic rigor
4. The text seems to be in a different font “where represents the spectral entropy, shows the power at the ÿý ÿÿ ÿ 2 ý/ bin of the power spectrum, and ý denotes the number of frequency bins.” Kindly check.
5. In the self-attention section, Q, K, and V should be properly defined and references should be provided.
6. The author may need to get assistance from a fluent English speaker to work on the overall grammar and structure of the paper.

Reviewer 2 ·

Basic reporting

The paper “Emotion detection from handwriting and drawing samples using an attention-based transformer model” is well presented and addresses a timely issue by providing solutions for emotion detection. However, there are areas that will benefit from minor refinement. I recommend accepting the article with minor edits. These suggested edits will contribute to the paper's overall quality and alignment with the journal's publication standards.

Experimental design

1. No experimental setup details are given. For example, how the hyperparameters were set.

Validity of the findings

1. The tables are well-organized but somewhat lacking in detail. The author should consider providing more comprehensive explanations for the tables to improve understanding.
2. Figure. 2 has no written details, why?
3. The same is the case with Table. 5 “comparison table”.
4. The author needs to add more written details to the tables in “Section 5” and “Section 5.1” to enhance the readability of the paper.

Additional comments

1. While the paper is well-presented, there are notable issues with sentence structure and grammar. The author may seek assistance from a fluent English speaker or Grammarly to address these concerns. E.g., Connect the sentence with the previous sentence “Electroencephalogram (EEG) signals from the user's brain are analyzed to determine their emotional state [6]”
--Also, restructuring “Expressions makes emotion detection from text a challenging problem [22]. Establishing clear and unbiased criteria for various states of emotional states becomes challenging due to individual handwriting styles may vary widely.” and others.

2. The number of citations is comparatively less as per the recent publication standards, I suggest adding more relevant citations can enhance the quality of the paper.

Reviewer 3 ·

Basic reporting

Article writeup is clear
Need English Language check as there are many grammatical mistakes Grammarly shows almost 300+ mistakes.
add more literature review & latest one.
Explain your methodology/algorithm in detail.

Experimental design

Explain your design & evaluation little bit more

Validity of the findings

Findings are clear
Table need deliberation

Additional comments

add appropriate references & more relevant keywords

---

## Round 0.2 · Minor Revisions

See comments from Reviewer 1.

Reviewer 1 ·

Basic reporting

no comment

Experimental design

no comment

Validity of the findings

no comment

Additional comments

Dear Editor:
I hope this message finds you well. The revisions aimed at improving clarity and overall quality have been implemented; however, I believe a few additional minor changes can further enhance the paper's rigor and readability.

1. The contribution can be presented either in bullet points (not highlighted) or as a single paragraph.
2. Ensure uniformity in the font of citation numbers; it should be either Calibri or Times New Roman.
3. Provide a title or number for the ED detection algorithm.
4. Adjust the font of Section 3.3.2 to Times New Roman to align with the rest of the paper.
5. Normalize the font size of Equation 8 to match the size of the rest of the equations.
6. In Table 5 (*Comparison...), better present citations in a regular manner, e.g., 9, 32, 38, 39, 40.

Reviewer 2 ·

Basic reporting

I am accepting this paper for publication as it meets our criteria for clarity and professionalism in language, provides adequate literature references and field context, adheres to standard article structure with well-presented figures, tables, and accessible raw data, and is self-contained with results relevant to the proposed hypotheses. The paper aligns with our standards of clarity, relevance, and academic rigor.

Experimental design

No Comments

Validity of the findings

No comments

Reviewer 3 ·

Basic reporting

authors have improved the article as per desired changes.

Experimental design

Aims & scope were addressed properly.

Validity of the findings

Validation of results were addressed properly.

Additional comments

authors have improved the article as per desired changes.

---

## Round 0.3 · accepted · Accept

The quality of the paper now meets the standard.

Reviewer 1 ·

Basic reporting

no comment

Experimental design

no comment

Validity of the findings

no comment

Additional comments

After reviewing the paper three times and ensuring all errors are corrected, it's now ready for publication. I agree with and accept the paper.